# A Study on the Path to the Sustainable Development of Sports-Consuming Cities—A Qualitative Comparative Analysis of Fuzzy Sets Based on Data from 35 Cities in China

**DOI:** 10.3390/ijerph191610188

**Published:** 2022-08-17

**Authors:** Xinze Li, Ronghui Yu, Chenjie Yan, Hongwei Xie

**Affiliations:** 1School of Physical Education, Hunan University of Science and Technology, Xiangtan 411100, China; 2School of Foreign Languages, Hunan University of Science and Technology, Xiangtan 411100, China; 3Physical Education School, Jimei University, Xiamen 361000, China

**Keywords:** sports consumption city, fuzzy-set qualitative comparative analysis method, sustainable development

## Abstract

The prospects of China’s sports sector hinge on how sports cities can thrive sustainably in the context of the new global pandemic, unlocking consumer potential and boosting domestic demand. In this study, 35 Chinese cities were chosen as research samples, and research methods such as literature, logical analysis, and fuzzy-set qualitative comparative analysis were used to select conditional variables such as government policy promotion and assistance, expert human resources, sports competitions and events, stadiums and facilities, and sponsorship by sports enterprises to examine how Chinese sports-consuming cities can develop sustainably. The research discovered that sports contests and events, as well as stadiums and facilities, are the essential prerequisites for the sustainable growth of sports-consuming cities, and that diverse combinations of the two may play a vital role in different circumstances. For the sustainable development of sports-consuming cities, there are four clusters and three models, which correspond to the “Venue + Event” model (Clusters 1 and 2), the “Event-led” model (Cluster 3), and the “Venue-led” model (Cluster 4). To encourage the high-quality growth of China’s sports business, each city may establish its development strategy based on its unique qualities. The goal is to supply Chinese expertise for the long-term growth of Western sports cities.

## 1. Introduction

The essential function of consumption in China’s economic development has grown in importance over the past several years due to China’s economic transformation and changes in international relations. Consumption in China will constitute 54.3 percent of the country’s GDP in 2020, exceeding capital creation, imports, and exports. It will also play a significant part in the double cycle, driving up consumption as an essential driver of urban economic growth. The general public now frequently chooses to spend and save money, and participation in sports is becoming more and more prevalent as an investment in health consumption. According to the China Urban Sports Consumption Report, sports consumption in China is estimated to be around 1.5 trillion and is expected to grow to 2.8 trillion by 2025, with a compound growth rate of over 13%. Sports consumption is expected to play a significant role in cities during this time. Sports consumption, as a significant component of the socioeconomic growth of contemporary cities, is crucial to raising cities’ international competitiveness, building up their cultural legacies, establishing their brand identities, and enhancing the general quality of life [1,2]. In addition to the “Opinions on Promoting National Fitness and Sports Consumption to Promote High-Quality Development of Sports Industry” policy, the General Administration of Sports of China has developed 40 pilot cities for sports consumption based on local natural resources. Building international sports towns has helped nations such as the UK, the US, and Germany revamp their economic and geographical systems, increasing their domestic and global importance [3,4]. Therefore, establishing a sports-consuming city is following China’s contemporary demands and is advantageous for the country’s national revitalization, health, urban development, and economic growth. It also has significant theoretical relevance and practical usefulness.

Sports consumption has developed as a critical social economy tool with the growth of social productivity, enhancing the scope and depth of sports cities. By supporting professional sports clubs and staging major sporting events in the latter part of the nineteenth century, Indianapolis, the capital of the US state of Indiana, quickly attracted significant amounts of social capital [4,5]. This helped to increase domestic demand for sports consumption and revitalize the city. The inclusion of classic industrial centers such as Manchester and Sheffield is similar [6,7]. Some academics contend that in the first half of the 20th century, local governments in traditional industrial cities in the United States started to renovate or invest in large stadiums to promote the sustainable development of sports consumption through professional sports, thereby promoting the revitalization of urban economies [8,9]. This action was taken to combat the issues of economic decline, rising unemployment, declining population, and vacant land. Such construction approaches can stimulate local economic growth, wide-ranging social benefits, and the mobilization and building of social capital. Additionally, well-known tourist destinations such as Barcelona, Sydney, and Athens have managed to redistribute economic activity in their cities due to a mix of significant athletic events and natural resources, which has helped fuel the expansion of sports consumption [10]. Chinese cities such as Beijing, Shanghai, and Hangzhou have increased their capacity for sports consumption at the same time by hosting several significant athletic events such as the Olympic Games, the Asian Games, and the Youth Olympic Games, intending to create high-quality sports towns [11,12,13].

Good and evil are mutually incompatible, and the process of developing a sports city will always have a negative influence on the growth of urban sports consumption. The “no city to host” Olympic Games is a classic negative effect. Affected by the new coronavirus pandemic, the Games will exert significant financial strain on local governments, inflict significant economic constraints on people’s lives, and pose substantial threats to the environment. At one point, Budapest’s “anti-Olympic community” accumulated 300,000 signatures, considerably above the legal limit of 138,000, prompting the city to withdraw from the bid. Furthermore, some academics feel that holding significant athletic events might lead to incidences of mob violence and possibly increased crime rates [14]. For example, when the Brazilian national team lost 1:7 to Germany in the World Cup in Brazil, it sparked riots in several Brazilian cities and a significant increase in crime, which had not only negative economic consequences for the cities’ development but also depleted social capital, which was highly detrimental to the increase in sports consumption [15].

Theoretical research on sports cities by domestic and international scholars has been fruitful. However, scholars have different perceptions of the connotations of sports cities due to differences in cultural heritage between China and the West. Differences in social resources and natural endowments of cities at home and abroad have led to the construction paths of sports cities at home and abroad having different characteristics. The globally recognized “Top Sports Cities in the World” index cannot be wholly integrated with China’s sports city development. The indications for the European Capital of Sport, for example, comprise five areas: “enjoyment of the sport, fulfillment of desires, feeling of community, learning to play fair, and improving health [16].” Sports city indicators in the United States include the importance of sport in a city’s visibility and the socioeconomic effect of professional sporting events on the city [17,18,19]. The indicators of China’s sports cities focus on both hardware and software, with hardware indicators focusing on the supply of basic sports facilities and supporting urban public facilities, and software indicators focusing on the diversity of sports culture, the flexibility of resource allocation, the activeness of popular sports, and social capital participation [20,21,22].

Scholars often theorize from sociological, planning, and economic viewpoints and commonly employ structural equation modeling, regression analysis, and system dynamics simulation models for quantitative analysis. However, the majority of available research focuses on the effect of a single or a few variables on sports cities, and few have been refined from the viewpoint of sports consumption, resulting in a lack of clarity on the influence mechanism of sports-consuming cities in China. Furthermore, various variables have a relative ‘net influence’ on sports cities, but combining these factors may not create the same outcomes, due to diverse action methods [23,24]. This is referred to as causal asymmetry, ‘in contrast to the previous sample symmetrical linear link between antecedent and consequent’.

This research aims to examine the various approaches to developing a sports-consuming city in China. As a result, this study employs the fsQCA technique and 35 pilot sports consumption cities in China as research examples to examine the development pathways of Chinese sports consumption cities. Because conventional qualitative and quantitative analyses have difficulties avoiding the causal complexity of describing the impact of varied combinations of variables on study findings, fsQCA was selected [25]. The fsQCA is a new research method that integrates qualitative and quantitative aspects, considers the results as a multivariate combination of different conditional variables, and finds the affiliation between different conditional variables using fuzzy analysis, simplifying “causal complexity [25].” Of course, this paper uses sports consumption as a starting point to investigate the interactive effects of factors such as “government policy promotion and assistance”, “expert human resources”, “sports competitions and events”, “sports venues and facilities’’ and sponsorship by sports enterprises on sports-consuming cities, demonstrating China’s multifaceted grouping of sports-consuming cities.

## 2. Theoretical Basis

Based on relevant research findings, this paper examines the complex causal relationship between sports consumption and the hardware and software-based construction of a sports-consuming city, taking into account factors such as “government policy promotion and assistance”, “expert human resources”, “sports events and activities”, “sports venues and facilities” and “sponsorship by sports enterprises” among others. Figure 1 depicts the theoretical model. It should be noted that the theoretical model does not cover all conceivable components but concentrates on the grouping influence of five factors on developing a sports-consuming metropolis.

### 2.1. Promotion and Assistance of Government Policies

China is a government-led nation, and the top-level design of the country dictates the growth path of sports-consuming cities. In terms of history, the State Council issued the Guidance Opinions on Accelerating the Development of the Sports Industry in 2010, signaling the beginning of China’s concentration on the development of the sports industry and starting official support for sports-consuming cities. In order to increase the size of the sports industry and establish a solid basis for creating sports consumption cities, China proposed the policy “Several Opinions on Accelerating the Development of the Sports Industry and Promoting Sports Consumption” in 2014. In 2019, China issued the Opinions on Promoting National Fitness and Sports Consumption to Promote the High-Quality Development of the Sports Industry, aiming to implement the national fitness strategy thoroughly, promote the development of mass sports, promote supply-side reform of the sports market, enhance the capacity of sports consumption, and improve the social and economic benefits. To mitigate the impact of the new coronavirus pandemic on the sports industry, China issued the Opinions on Further Releasing Consumption Potential to Promote Sustainable Consumption in 2022 to accelerate the construction of a new development pattern, making concerted efforts and striking a balance between far and near, adopting a comprehensive approach to release consumption potential, and promoting sustainable recovery of sports consumption. Furthermore, the People’s Republic of China’s Sports Law has been regularly revised to protect the public’s right to fitness, regulate the sports market, and support the promotion and execution of government programs. Examples include
Tax benefits for sports facility repair.Specific financing assistance for sporting event marketing.Scientific direction for the growth of mass sports.

In total, China has 4 administrative laws and 16 State Council and central government documents on the sports industry, which are intended to improve the quality of development of the sports industry, increase consumption potential and international influence, and provide strong support for the development of sports consumer cities.

### 2.2. Expert Human Resources

Creating high-quality expert human resources is a critical step in promoting the long-term growth of sports-consuming cities. On the one hand, the growth of sports cities necessitates the development of longitudinal talent, which is favorable in breaking the conventional cognitive system, encouraging sports cross-fertilization with other disciplines, and enhancing the depth of sports industry development [26]. On the other hand, the development of sports cities necessitates the mobilization and accumulation of multi-level social capital, giving full expression to social capital’s critical role in allocating sports resources and providing breadth for the growth of sports businesses [27]. On the other hand, an excellent human environment encourages the development of skilled human resources, and the two are mutually reinforcing. The establishment of an “industry–research–academia” cooperation model between schools and enterprises accelerates student socialization, promotes diverse student development, facilitates interaction between professional knowledge and practical skills, aids in bridging the micro gap between school curricula and market demand, and aids in improving the quality of public sports service provision [28]. It is worth noting that by outsourcing public sports services to social sports organizations, the government has activated the market’s competitive “survival of the fittest” mechanism, improved the quality of professional human resources, met the diverse sports needs of the public and market, released the potential of sports consumption, and promoted the development of sports cities.

### 2.3. Sports Competitions and Activities

Sports events are crucial to the development of a sports-consuming city. Hosting athletic events in a city will considerably raise the city’s profile, promote public excitement for the sport, release the potential for sports consumption, optimize the city’s supply structure, and improve social cohesion. The Winter Olympics and Paralympic Games will be hosted in Beijing in 2022, making it the world’s most renowned ‘City of Two Olympics’ and garnering worldwide attention. On the one hand, marketing a series of branded items such as “Ice Dundun” and “Snow Rong Rong” has unleashed the potential of sports consumption while promoting regional economic growth. The national trend of “ice and snow fever,” on the other hand, has promoted the comprehensive development of ice and snow sports, effectively fulfilling the strategic goal of “300 million people taking to ice and snow,” promoting the sustained recovery of China’s sports consumption, and advancing the in-depth development of national fitness. Thus, athletic activities increase the quality of sports facility supply, create economic advantages, and stimulate the mobilization and accumulation of social capital. Most studies have found that sporting events have become a focal point of the city’s economy and culture and have promoted the long-term development of sports consumption through changes in operating philosophy, optimization of sporting structural relationships, and increased profitability and stability [29]. Wang Yongshun believes that national fitness events can not only meet the health consumption needs of the masses and promote the accumulation of social capital but also improve social acceptance, effectively reduce the crime rate, and maintain social stability. Finally, athletic events are a tangible reflection of the city’s legacy and a necessary means of encouraging sports consumption.

### 2.4. Sports Venues and Facilities

By 2019, China’s stadiums will have grown by 34% since 2013, setting the groundwork for developing sports-consuming cities. According to the “Outline for the Construction of a Strong Sports Nation,” stadiums must help the sports industry become a pillar industry, actively implement existing stadium land and tax policies, optimize the land supply model, and strengthen the spatial planning of the “golden edge and silver corner.” Some academics argue that stadiums that fulfill international sports event criteria benefit municipal brand marketing and the promotion of social and economic development if they successfully host significant international sporting events. The “Water Cube” stadium, for example, is China’s iconic Olympic legacy, serving as a swimming pool during the Summer Olympics, a curling pool during the Winter Olympics, and a water park at standard times, providing a significant contribution to the area’s sports economy’s long-term growth [30]. This demonstrates that diversifying the roles of sports arenas is critical to increasing sports consumption. Stadiums are classified into three types based on their governance: public, school, and commercial. In sports, public stadiums primarily serve the crowds. These stadiums are intended to satisfy the population’s requirements, beginning with their health and participation in sports. However, public sports facilities are hampered by a lack of sports land and a lopsided distribution in spatial planning, leading to inadequate supply capacity. School sports halls are primarily used for school and competitive sports. They are utilized mainly by students and professors and are less accessible and more exclusive to the general public. Commercial stadiums are mainly intended to address the public’s deep-seated athletic requirements, compensate for the failure of public stadiums to meet them, and boost the public’s endogenous motivation. Furthermore, involvement in sports not only enhances the quality of life but also accumulates social capital, and sports facilities directly impact participation in sports. Sports venue stratification may therefore be more effective in boosting participation and generating demand for sports consumption [31,32].

### 2.5. Sponsorship of Sports Enterprises

Sustainable growth of sports-consuming cities needs multi-level and multi-dimensional assistance, with social capital increasingly becoming essential in fostering the qualitative development of the sports business. According to Ren Bo and others, sports firms thrive under China’s new “two-cycle” growth pattern. Backbone firms exhibit the “strongest is always stronger” horse effect, while small, medium, and micro companies rapidly expand into an active element of the sports sector. Furthermore, sports social organizations in China have increased by 108 percent over the previous six years (2012–2017), a remarkable statistic indicating that China has steadily completed the accumulation of social capital in sports. Companies may maximize their advantages by sponsoring significant athletic events, which will boost their economic rewards, exposure, and brand image and deepen customers’ perceptions of their goods [33]. Red Bull, for example, solely supports extreme sports, resulting in a win–win scenario for both the sports event and the corporation. Companies have considerably increased the commercial and social value of sports by supporting sports clubs, social organizations, and individual athletes, promoting the long-term growth of sports, revitalizing the sports market, and stimulating domestic demand for sports consumption. Adidas [34,35], for example, sponsored Manchester United FC, whereas 361° supported Sun Yang.

## 3. Study Design

### 3.1. Selection of Research Methods

This research uses QCA to investigate the pathways for developing sports-consuming cities in China. Created in 1987 by Ragin, the technique is based on set theory and Boolean operations and aims to establish overarching causal linkages from identical or different situations, exposing the many channels via which social events emerge. Compared to classical regression analysis, QCA provides the following advantages: 1. It is appropriate for small and medium sample numbers; 2. There is equality between the fitted production routes, shattering the ideal solution attitude and 3. reducing multiple covariance interference; 4. The combined form of the condition variables is examined from a systems perspective, reducing the direct influence of single factors and highlighting the causes and mechanisms of complexity. The paper’s intricacy in analyzing the building pathways of sports-consuming cities in China and its worth in providing a cross-reference for other geographic locations make QCA an ideal choice [36,37,38,39].

Depending on the set form, QCA is further categorized as csQCA, mvQCA, and fsQCA, wherein fsQCA is a more sophisticated version of csQCA. The csQCA has statistical limitations since it can only handle conditional and outcome variables with values ranging from 0 to 1. The idea of affiliation is introduced in fsQCA, which changes the connection between conditions and outcomes in a degree relationship. The original data may be “calibrated” to a score of 0 to 1 by establishing a cut-off line and then calculating consistency and coverage using Boolean techniques to assess the requirement and suitability of various combinations of circumstances leading to outcomes [40,41]. Of course, none of the five condition variables listed in the preceding section can be measured, and the traditional method of determining “yes or no’’ lacks a certain degree of scientific validity, so it is clear that categorizing condition variables into levels based on scientific evidence is closer to the truth. In this research, fsQCA 3.0 software (https://www.file-extensions.org/fsqca-file-extensions) was used to design the analytic framework based on the sample, construct the variables and assignment rules, calibrate and generate the truth table, and finish the path identification and interpretation of the findings (see Figure 2).

### 3.2. Study Case Selection and Data Source

In this study, 35 Chinese cities were chosen as the samples for two reasons: first, they were the first pilot cities for sports consumption in China, making them representative; second, the samples were chosen because they were located in different geographical locations in China, making them geographically specific.

The data for the outcome variables came from the 14th Five-Year Sports Plan and the Resident Sports Consumption Survey and Analysis Report for each Chinese city. The data for the conditional variables came from the Work Programme for the Construction of Pilot Cities for Sports Consumption in China and the Implementation Programme for Promoting National Fitness and Sports Consumption in Chinese Cities to Promote the High-Quality Development of the Sports Industry.

### 3.3. Variable Selection and Assignment Rules

In fsQCA, a value of [0, 1] is assigned to all samples, where “1” indicates associated and “0” means not connected at all, and higher values between “0” and “1” indicate more affiliation, making the findings more interpretable. All variables in QCA analysis are separated into outcome and conditional variables, corresponding to the dependent and independent or explanatory variables in classical regression analysis [42]. Table 1 shows the principles for assigning values to variables based on their meanings and measurements.

Based on the measurement mentioned in earlier criteria and assignment procedures, this study compiles the necessary information for each sports consumption city and assigns [0, 1] to the sample, obtaining the truth table (partially), as shown in Table 2.

## 4. Study Results and Analysis

### 4.1. Analysis of the Necessary Conditions

Before the conditional grouping analysis, a single conditional variable must be checked for consistency and coverage using the fsQCA procedures. The percentage of the samples with this condition that obtained the reported outcome is shown by consistency, and coverage denotes the percentage of all samples that met the given result that had this condition. When the consistency is more prominent than 0.9, the condition variable is necessary for the result variable [24,39]. Table 3 displays the findings of the fsQCA necessity analysis, where the consistency indicators for “sports games and events” and “sports venues and facilities” are more significant than 0.9, indicating that “sports games and events” and “sports venues and facilities” should be included Table 3 displays the findings of the fsQCA necessity analysis.

### 4.2. Conditional Configuration Analysis

This research analyzes data from 35 sports-consuming cities in China using fsQCA 3.0 software to provide three solutions of varying complexity: a complicated solution, a parsimonious solution, and an intermediate solution. The intermediate answer is often seen to be more compatible with empirical data and so the best interpretative option for testing, building, and improving the theory. In most research studies on QCA approaches, intermediate solutions are employed [40]. Furthermore, because the simplex solution incorporates all logical residuals, antecedent configurations included in both the simplex and intermediate solutions are referred to as core conditions. In contrast, antecedent configurations that are only included in the intermediate solution are referred to as marginal conditions [25,41].

By examining the criteria in the intermediate and parsimonious solutions, four group states with consistency larger than 0.8 were identified (e.g., Table 4). The overall consistency of 0.873823 suggests that 87.4 percent of sports-consuming cities that fall into one of these four categories are highly developed; the overall coverage was 0.601296, suggesting that these four conditional groups explained 60.1 percent of the city sample’s sports consumption. The answers’ consistency and coverage surpassed the critical levels, confirming the validity of the empirical study.

Grouping 1 has a consistency of 0.73 and coverage of 0.67 for “Promotion and assistance of government policies × sports contests and events × sports venues and facilities” (where “and” means “not,” the same as below). The fundamental factors were sports tournaments and events, as well as sports venues and infrastructure, whereas the marginal conditions were government promotion and aid.

Grouping 2 (“Expert human resources × sports contests and events × sports venues and facilities”) has a consistency of 0.62 and coverage of 0.74. Here, sports contests and events, as well as sports venues and infrastructure, are essential criteria, whereas specialist human resources are peripheral.

Grouping 3 (“promotion and assistance of government policies × expert human resources × sports contests and events × sponsorship of sports companies”) has a consistency of 0.74 and coverage of 0.43. Here, sports tournaments and events were the primary criteria, with government policy promotion and aid, specialist human resources, and sponsorship of sports firms serving as secondary factors.

Grouping 4 (“government policy promotion and assistance × expert human resources × stadiums and facilities × sports enterprise sponsorship”) has a consistency of 0.63 and coverage of 0.51. Here, the primary prerequisite was stadiums and facilities, with government policy promotion and aid, expert human resources, and sports business sponsorship as secondary criteria.

### 4.3. Robustness Test

Using the findings of previous investigations, this study performed robustness checks using data calibration. The result variable’s three-valued fuzzy set was recalibrated by lowering the complete affiliation from 0.95 to 0.80 and the full disaffiliation from 0.05 to 0.20, with no modifications in the other stages. The conditional grouping analysis demonstrated that the study’s primary conditions, grouping pathways, consistency criteria, and case frequencies did not vary much, suggesting that the findings were robust.

### 4.4. Case Analysis

This research divides the building trajectories of sports-consuming cities into three models based on the four groups.

The “Venue + Event” concept corresponds to Groupings 1 and 2, representing the cities of Xiamen and Ningbo, respectively. Xiamen and Ningbo, China’s fastest expanding new tier-1 cities, have a solid economic foundation. In terms of sports venues and facilities, (1) Xiamen has strengthened the intelligent construction of public sports venues through the “PPP” approach, improving the quality of public service provision; (2) Ningbo has cleverly used spatial structure to revitalize the stock of sports facilities and solve the issue of fitness difficulties for the public; (3) Xiamen has continued to promote the integration of sports venues with businesses, green areas, and parks, resolving the problem of mass fitness difficulties. Both give considerable subsidies to attract top-level sporting events with the great societal appeal, a broad market area, and strong growth possibilities, such as the International Marathon and the Asian Games. Furthermore, Xiamen has used its port to connect with towns such as Quanzhou and Zhangzhou to organize international sporting goods fairs, thus boosting the scope of the sports market and strengthening the city’s worldwide importance. Ningbo aggressively encourages “business, research, and academic” growth, continually enhances talent subsidies, and attracts high-quality expert human resources. This model is suited for contemporary cities with a well-developed industrial structure and significant social capital and concentrates on the development of stadiums and the staging of athletic events, with a strong public demand for popular and professional sports and a large capacity for sports consumption.

The city of Xinyu represents the “event-led” approach, which corresponds to configuration 3. Xinyu is a well-known tourist destination in China, not only for its natural beauty but also for its extensive recreational facilities. Regarding professional sports events, Xinyu has fully used Xianniu Lake’s unique natural geographical advantages to aggressively conduct elite international bicycle races, propelling the area’s economy’s growth. In terms of mass sports events, Xinyu actively promotes fitness qigong, gateball, square dance, and other sports activities, which are firmly integrated with the benefits of its natural oxygen bar, allowing the masses to create a healthy body and mind. Furthermore, leisure and wellness facilities allow athletes, the elderly, and other groups to relax, recuperate, and socialize, increasing the bar for secondary consumption. This model is suited for recreational towns with a well-developed “green + events + recreation” system and good natural assets since it concentrates on establishing high-quality sporting events with a strong need for recreation and rehabilitation.

The city of Zhangjiakou illustrates the “venue-led” paradigm in configuration 4. Zhangjiakou is a typical Chinese industrial city, but its poor natural environment and unique geographical position have hampered its economic growth, and it was designated as a “Beijing-Tianjin Poverty Belt” by the Asian Development Bank in 2005. Zhangjiakou’s economy has soared since its successful bid for the Olympic Games in 2015, keeping with the idea of sport for development. Zhangjiakou has converted underused industrial space into usable sports space, encouraging the sports industry’s long-term growth. The Nordic Centre cross-country ski area, the Nordic Centre ski jumping area, the Biathlon Centre, and Sports Ski Park sites A and B are only a few examples. In terms of mass sports, Zhangjiakou’s ski slopes offer a unique natural endowment that stimulates the desire and potential for mass sports consumption while promoting economic revitalization. Zhangjiakou’s stadiums fulfill the demands of professional athletes in training, and the sportspeople often follow behind renowned sports teams, bringing a vital human resource of specialists to the city. Furthermore, Zhangjiakou City has developed the “Torch” program for the Winter Olympic potential, which has helped to accumulate social capital. This strategy focuses on building stadiums in areas with a high public demand for leisure and professional sports, and it is appropriate for industrial towns with abundant natural resources but poor socioeconomic performance.

## 5. Conclusions, Suggestions and Prospects

### 5.1. Conclusions

Against the background of the new global pandemic, the sports industry has been heavily damaged, and how sports-consuming towns may further expand and promote economic advantages is discussed. Studies have shown that government policy promotion and assistance, expert human resources, sporting events and activities, stadiums and facilities, and sports business sponsorship can be used as single variables to influence the sustainable development of sports-consuming cities. However, no study has yet discussed how these five variables synergistically contribute to sports-consuming cities’ development. As a result, this research analyses the building trajectories of sports-consuming cities using fsQCA on a sample of 35 Chinese cities based on the Chinese context. This paper’s results are twofold.

The first point is that sports events and activities, as well as stadiums and infrastructure, are all required for the long-term growth of a sports-consuming city.

The second argument is that there are four groups and three models for the sustainable development of sports-consuming cities. The “Venue + Event” model: the core conditions are sports events and activities, stadiums, and facilities, which are appropriate for modern cities with a complete industrial structure and substantial social capital. The “event-led” paradigm is appropriate for recreational communities with a well-developed service structure and abundant natural resources. “Venue-led” model: sporting venues and facilities are the primary conditions appropriate for industrial towns with favorable natural circumstances and socioeconomic depression.

### 5.2. Suggestions

The study’s findings indicate that the sustainable growth of Chinese sports-consuming cities is skewed toward establishing hardware facilities while overlooking the influence of software facilities. Based on this, the research makes the three suggestions listed below.
(1)Actively encourage the revision of the People’s Republic of China Sports Law and the establishment of a sound system of rules and regulations to monitor the execution of sports policies, control the order of the sports market, and defend the public’s rights and interests in sports.(2)Strategic planning and development tailored to local circumstances. Different combinations of factors result in various cause–effect correlations. Each city’s geographical location, resource endowment, economic level, cultural legacy, and policy orientation vary, resulting in diverse growth pathways for sports-consuming communities. The government should appraise the city’s assets in terms of hardware and software and create a route for developing a sports-consuming city based on local circumstances.(3)Strengthening the hardware on the longboard Stadiums and athletic events is a critical component of the long-term growth of sports-consuming cities. First, vigorously promote the PPP model to enhance the quality of public sports service provision. Second, make information on sports land publicly accessible and calculate the extent of land taxes. Finally, we will foster the long-term growth of the sports business by expanding on the notion of “carbon neutrality.”(4)Make up for software flaws. To begin, extend the paradigm of collaboration between business, research, and universities to cultivate excellent sports talent consistently. Second, the Chinese government should take the lead in improving governance effectiveness and promoting the execution of sports policies. Finally, the market competition mechanism is optimized to encourage social capital formation.

### 5.3. Outlook and Shortcomings

Shortcomings include the following: first, the theoretical model only comprises five conditional variables, and the explanations have yet to be further substantiated. Second, the research was confined to China and did not include overseas samples; therefore, cross-sectional comparisons were absent. The examination of the circumstances grouping does not entirely represent “sponsorship by sports corporations,” most likely because the sports industry has been struck by the new coronavirus pandemic, resulting in low social capital.

Prospects: the addition of more robust explanatory indicators, as well as the incorporation of foreign samples, will offer a more intuitive indication of the numerous concurrent causal links in sports cities, as well as enhance their scientific growth.

## Figures and Tables

**Figure 1 ijerph-19-10188-f001:**
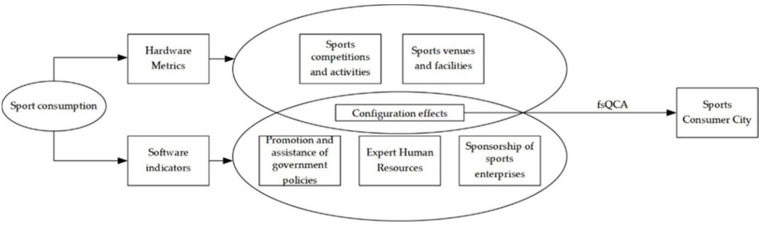
Theoretical model for sustainable development of sports-consuming cities in China.

**Figure 2 ijerph-19-10188-f002:**
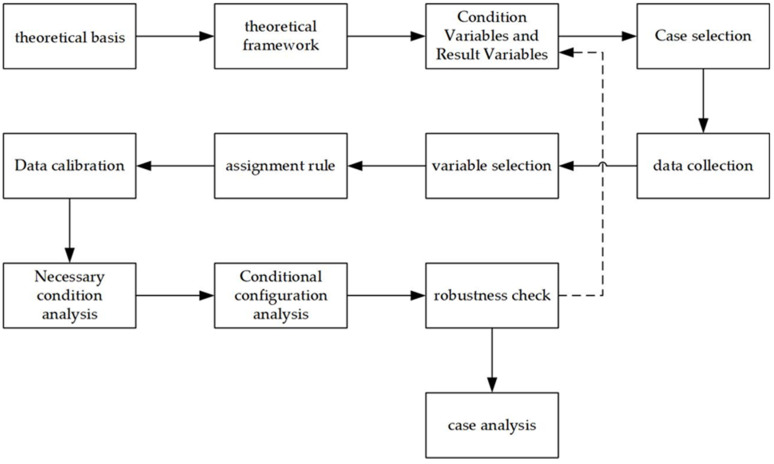
Flowchart of fsQCA.

**Table 1 ijerph-19-10188-t001:** Sustainable growth of sports-consuming cities: Variable interpretation, assignment guidelines, and data sources.

Variable Category	Variable Name	Variable Explanation	Metrics	Assignment Rules	Data Sources
Result variables	Sustainable development of sports-consuming cities	Assessing the potential for high-quality development of the urban sports industry	The total scale of sports consumption in various cities in 2020	Calibrate according to ternary fuzzy set (0.95, 0.5, 0.05)	Each city’s “14th Five-Year Plan for Sports Business” and “Resident Sports Consumption Survey and Analysis Report” in 2020
Condition variable	Promotion and assistance of government policies	Provide guarantees and opportunities for the high-quality development of the sports industry	a. Provide preferential policies for the development of the sports industry; b. Create a resource trading platform for the sports industry; c. Implement the “Sports+” model; d. Optimize the supply mechanism of public sports services; e. Serve or invite a certain Head of international sports organizations; f. Assist with financial services	1 for both; 0.67 for more than 4 items; 0.33 for more than 2 items; 0 for none	The “Work Plan for the Construction of China’s Sports Consumption Pilot Cities” and the “Implementation Plan for Promoting National Fitness and Sports Consumption to Promote High-Quality Development of the Sports Industry” issued by various cities
	Expert human resources	Provide high-quality composite talents for high-quality sports industry development	a. Improve the social sports talent archive; b. Improve the professional sports certification system; c. Use schools, enterprises, or social channels to cultivate professional talents; d. The government, schools, and social organizations link up with each other to form a cooperative relationship between industry, education, and research	1 for both; 0.75 for 3 items; 0.5 for 2 items; 0.25 for 1 item; 0 for none	
	Sports competitions and activities	Build a solid foundation for the high-quality development of the sports business	a. Hold national sports events; b. Hold international sports events; c. Hold a professional competition; d. Host a national fitness event	All meet 1; meet 2 items for 0.67; meet 1 item is 0.33; both are not satisfied with 0	
	Stadiums and facilities	Provide a stable material guarantee for the high-quality development of the sports industry	a. Optimize the land supply of sports venues; b. Improve the construction of sports venues (intelligent); c. The operating mechanism of innovative sports venues (including OT, ROT, and BOT)	All is satisfied with 1; 0.67 for 2 items; 0.33 for 1 item; 0.33 for 1 item; all are not satisfied with 0	
	Sponsorship of sports enterprises	Provide economic support for the high-quality development of the sports industry	a. Stadiums receive various amounts of corporate support.; b. Various levels of corporate funding for sporting events; c. Companies provide multi-level support for athletes	All is satisfied with 1; 0.67 for 2 items; 0.33 for 1 item; and 0.33 for 1 item; and 0 for none	

**Table 2 ijerph-19-10188-t002:** The truth table for the long-term growth of sports-consuming cities (partial).

City (Autonomous Region, Municipality Directly under the Central Government)	The Construction of Sports Consumption City	Promotion and Assistance of Government Policies	Specialist Human Resources	Sports Competitions and Events Stadiums	Facilities Sports	Corporate Sponsorship
Ningbo City, Zhejiang Province	0.79	0.67	0.75	1	1	0.67
Shaoxing City, Zhejiang Province	0.52	0.33	0.75	1	1	0.33
Jinhua City, Zhejiang Province	0.47	0.33	0.75	1	0.67	0.33
Hefei City, Anhui Province	0.77	0	0.25	1	0.67	0.33
Huangshan City, Anhui Province	0.05	1	0.5	0.67	0.67	0.33
Fuzhou City, Fujian Province	0.74	1	0.5	1	1	1
Xiamen City, Fujian Province	0.59	1	0.25	1	0.67	1
Sanming City, Fujian Province	0.09	0	0.75	0.67	0.67	0.67
Nanchang, Jiangxi Province	0.5	0.33	0.5	1	0.67	0.33
Xinyu City, Jiangxi Province	0.05	0.33	0.25	0.67	0.67	0.33

**Table 3 ijerph-19-10188-t003:** An examination of the need for the long-term growth of sports-consuming cities.

Variable	Consistency	Coverage
Promotion and assistance of government policies	0.666667	0.679463
~Promotion and assistance of government policies	0.622097	0.511616
Expert human resources	0.770873	0.566682
~Expert human resources	0.563088	0.672918
Sports competitions and activities	0.979912	0.508635
~Sports competitions and activities	0.135593	0.50116
Sports venues and facilities	0.915882	0.526145
~Sports venues and facilities	0.33145	0.726272
Sports enterprise sponsorship	0.691149	0.636784
~Sports enterprise sponsorship	0.673572	0.605872

**Table 4 ijerph-19-10188-t004:** A study of the long-term growth of sports consumer cities.

Condition	Configuration 1	Configuration 2	Configuration 3	Configuration 4
Promotion and assistance of government policies	•		○	○
Expert Human Resources		•	○	•
Sports Games and Events	●	●	●	
Sports venues and facilities	●	●		●
Sports corporate sponsorship			○	○
Consistency	0.72541	0.623219	0.74026	0.626374
Original coverage	0.666667	0.741369	0.429379	0.500942
Unique coverage	0.0734463	0.042059	0.0313873	0.0188321
Total consistency	0.873823			
Total coverage	0.601296			

Note: “●” indicates that the core condition exists, “○” indicates that the condition does not exist, “•” indicates that the edge condition exists, and “blank” indicates that the condition is at a medium level.

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
