# Peer review of "A Study on the Path to the Sustainable Development of Sports-Consuming Cities—A Qualitative Comparative Analysis of Fuzzy Sets Based on Data from 35 Cities in China"

_ijerph, 2022, doi:10.3390/ijerph191610188_

Round 1

Reviewer 1 Report

The work is devoted to the classification of Chinese cities associated with the development of physical culture and sports. The authors tried to classify cities according to the level of development in the field of sports and identify factors that contribute to the further development of these cities and the improvement of infrastructure.

The originality of the work is 93.34% in the "Antiplagiarism" system, which is a very good indicator. The article is well-written, there are tables. The article consists of sections: introduction, theoretical base, choice of data source, analysis of the obtained data, conclusions and recommendations. However, the shortcomings of the work should be noted:

1. Weak scientific research.

2. Research methods are practically not described.

3. The purpose of the work is not clear.

4. What was the hypothesis?

5. In the list of references, sources No. 7 and No. 26 are framed incorrectly.

6. Literary source No. 21 is not related to the work.

7. The fsQCA3.0 and MADM methods are not described in detail in the articles and are incomprehensible.

I recommend to significantly improve the article. It should have a more serious evidence base, improved mathematical statistics and visualization in the form of graphs or diagrams.

Author Response

List of responses

Dear Editors and Reviewers

Thank you for your letter and for the reviewers' comments concerning our manuscript entitled "A study on the path of sustainable development of sport-consuming cities - A qualitative comparative analysis of fuzzy sets based on data from 35 cities in China" (ID:ijerph-1837847). Those comments are all valuable and very helpful for revising and improving our paper, as well as the essential guiding significance to our research. We have studied the comments carefully and made a correction that we hope meets with approval. Revised portions are marked in red on the paper. The significant corrections in the paper and the responses to the reviewer's comments are as follows:

Responds to the reviewer's comments:

Reviewer #1:

  1. Response to comment: Weak scientific research.

Response: As you point out, the theoretical foundation of this paper is shaky. To make it more explanatory, the logic of the first part (introduction) and the second part (theoretical foundations) has been reorganized, and relevant theoretical arguments have been added.

  1. Response to comment: Research methods are practically not described.

Response: This was an error on our side, for which we sincerely apologize. This page draws on other articles and reorganizes section 3.1 (Selection of research techniques) to make it more accessible and operational so that future readers may refer to this approach in a logical manner.

  1. Response to comment: The work's purpose is unclear.

Response: The lack of clarity regarding the study's goal is a significant issue, which we have explicitly remedied in light of your concerns and underlined in Part I. (Introduction). The first section provides background information for the study, the significance of the investigation, research questions, and the study's objective.

  1. Response to comment: What was the hypothesis?

Response: Sorry if I misunderstood what you were trying to express with this comment, but I did my best to comprehend it and reflect it in the essay. Theoretical assumptions concerning a single component are required for regression analysis and structural equation modeling. Unlike traditional research methodologies, fsQCA focuses on the impact of several conditional variable combinations on outcome variables and does not need theoretical assumptions about a single component. If you are talking about assumptions regarding the study's conclusion, I have put this at the start of Part 2 (Theoretical Foundations) and drawn the theoretical model to make it simpler to grasp.

  1. Response to comment:In the list of references, sources No. 7 and No. 26 are framed incorrectly.

Response: We apologize for the error caused by a lack of attention on our side. We have updated the formatting of references Nos. 7 and 26 in light of your feedback.

  1. Response to comment: Literary source No. 21 is unrelated to work.

Response: Regarding your remarks, we have deleted reference number 21 and re-provided the literature foundation.

  1. Response to comment:The fsQCA3.0 and MADM methods are incomprehensible and not described in detail in the articles.

Response: In response to your feedback, we rewrote this critical part. The research abandons the MADM technique in favor of fsQCA. The differences between fsQCA and conventional analysis are briefly presented in the first part (introduction); in the third part, the contrasts between fsQCA and traditional analysis are underlined, and a flow chart of fsQCA is constructed to make it more approachable.

Special thanks to you for your good comments.

Other changes:

1.Section 4.4 (case studies) has been added to this to give in-depth case studies of various configurations to make them more representative and to provide theoretical references for other geographies.

2.The title, abstract, and keywords have been reorganized to give greater focus and clarity to the topic.

3.The format and content of the references have been reorganized to give the article more explanatory strength.

4.To make the article more understandable to reviewers and readers, the grammatical structure has been revised and touched up in its entirety.

We did our most complicated to enhance the text and made some adjustments. These modifications will not affect the paper's content or structure. Furthermore, instead of listing the modifications, we highlighted them in red in the amended document. We sincerely appreciate the editors/reviewers' dedicated effort and hope any modifications will be addressed. Thank you for your comments and recommendations once again.

Yours sincerely

Xinze Li

Reviewer 2 Report

The bits that I understood, I found interesting, informative and thought-provoking. I agree with your observation that much of the existing work on sport cities and sport event legacy has been conducted with a western lens, but am unsure how your Eastern lens differed, given that you were essentially looking at the same elements (tangible infrastructure and intangible social capital) and drawing the same conclusions as the existing research. the work is also lacking a critical lens, in terms of all the known failings and negative impacts attached to hosting sport event. 

Unfortunately, I failed to fully understand and follow large parts of it due to the language used and the poor grammatical construction of the sentences. the translation into English needs more work. I am also not familiar with the methodology used and have not seen it applied in this context before. it would appear to have potential, but needs explaining more clearly and concisely if others are to replicate what you have done.

i think that this has potential, but it needs to be re-written and re-structured in a manner that makes it easier to follow and to consume.

Author Response

List of responses

Dear Editors and Reviewers

Thank you for your letter and for the reviewers' comments concerning our manuscript entitled "A study on the path of sustainable development of sport-consuming cities - A qualitative comparative analysis of fuzzy sets based on data from 35 cities in China" (ID:ijerph-1837847). Those comments are all valuable and very helpful for revising and improving our paper, as well as the essential guiding significance to our research. We have studied the comments carefully and made a correction that we hope meets with approval. Revised portions are marked in red on the paper. The significant corrections in the paper and the responses to the reviewer's comments are as follows:

Responds to the reviewer's comments:

Reviewer #2:

  1. Response to comment:I found the bits I understood interesting, informative, and thought-provoking. I agree with your observation that much of the existing work on sports cities and sports event legacy has been conducted with a western lens. However, I am unsure how your Eastern lens differed, given that you were essentially looking at the same elements (tangible infrastructure and intangible social capital) and drawing the same conclusions as the existing research. The work also lacks a critical lens regarding all the known failings and negative impacts of hosting sports events.

Response: Thank you for your excellent recommendations; adjustments have been made in response to your remarks. It has also been detailed in the first section (Introduction) with information on the detrimental consequences of sports cities on the development process.

  1. Response to comment:Unfortunately, I failed to fully understand and follow large parts of it due to the language used and the poor grammatical construction of the sentences. The translation into English needs more work, and I am also unfamiliar with the methodology and have not seen it applied in this context before. It would appear to have potential but needs explaining more clearly and concisely if others are to replicate what you have done.

Response: I apologize for the flaws in our writing that hindered you from fully comprehending this post. On the one hand, we have drastically altered this article's grammatical structure to make it more intelligible. However, we have considerably altered the third part to offer a full explanation of the application of fsQCA in this article and created a flowchart of fsQCA to make it more explanatory for following readers to refer to and implement.

Special thanks to you for your good comments.

Other changes:

1.Section 4.4 (case studies) has been added to this to give in-depth case studies of various configurations to make them more representative and to provide theoretical references for other geographies.

2.The title, abstract, and keywords have been reorganized to give greater focus and clarity to the topic.

3.The format and content of the references have been reorganized to give the article more explanatory strength.

4.To make the article more understandable to reviewers and readers, the grammatical structure has been revised and touched up in its entirety.

We did our most complicated to enhance the text and made some adjustments. These modifications will not affect the paper's content or structure. Furthermore, instead of listing the modifications, we highlighted them in red in the amended document. We sincerely appreciate the editors/reviewers' dedicated effort and hope any modifications will be addressed. Thank you for your comments and recommendations once again.

Yours sincerely

Xinze Li